# Characterization of Mechanical and Cellular Effects of Rhythmic Vertical Vibrations on Adherent Cell Cultures

**DOI:** 10.3390/bioengineering10070811

**Published:** 2023-07-06

**Authors:** Dongho Kwak, Thomas Combriat, Alexander Refsum Jensenius, Petter Angell Olsen

**Affiliations:** 1RITMO Centre for Interdisciplinary Studies in Rhythm, Time and Motion, Department of Musicology, University of Oslo, 0371 Oslo, Norway; a.r.jensenius@imv.uio.no; 2Hybrid Technology Hub, Centre for Organ on a Chip-Technology, Institute of Basic Medical Sciences, University of Oslo, 0372 Oslo, Norway; t.m.d.combriat@fys.uio.no (T.C.); peteraol@medisin.uio.no (P.A.O.); 3Department of Physics, Njord Center, University of Oslo, 0316 Oslo, Norway; 4Unit for Cell Signaling, Department of Immunology and Transfusion Medicine, Oslo University Hospital, 0372 Oslo, Norway

**Keywords:** rhythmic vertical vibration, sound vibrations, Max/MSP signal generation, mechanobiology, F-actin filament length, F-actin filament thickness, F-actin filament angle, particle tracking velocimetry (PTV), acceleration, shear stress, image feature extraction

## Abstract

This paper presents an innovative experimental setup that employs the principles of audio technology to subject adherent cells to rhythmic vertical vibrations. We employ a novel approach that combines three-axis acceleration measurements and particle tracking velocimetry to evaluate the setup’s performance. This allows us to estimate crucial parameters such as root mean square acceleration, fluid flow patterns, and shear stress generated within the cell culture wells when subjected to various vibration types. The experimental conditions consisted of four vibrational modes: No Vibration, Continuous Vibration, Regular Pulse, and Variable Pulse. To evaluate the effects on cells, we utilized fluorescence microscopy and a customized feature extraction algorithm to analyze the F-actin filament structures. Our findings indicate a consistent trend across all vibrated cell cultures, revealing a reduction in size and altered orientation (2D angle) of the filaments. Furthermore, we observed cell accumulations in the G1 cell cycle phase in cells treated with Continuous Vibration and Regular Pulse. Our results demonstrate a negative correlation between the magnitude of mechanical stimuli and the size of F-actin filaments, as well as a positive correlation with the accumulations of cells in the G1 phase of the cell cycle. By unraveling these analyses, this study paves the way for future investigations and provides a compelling framework for comprehending the intricate cellular responses to rhythmic mechanical stimulation.

## 1. Introduction

Mechanical cues within the cellular microenvironment play a crucial role in influencing fundamental processes such as proliferation, migration, and differentiation [1,2,3]. Cells can perceive and convert mechanical stimulation into biochemical signals through mechanosensors, which include ion channels, cell adhesion sites, and the cytoskeleton [4].

In order to mimic a dynamic mechanical environment, numerous strategies for vibrating cell cultures have been developed [2]. Accordingly, vibration has been investigated across diverse cell culture models, such as fibroblast cells [1,5] bone cells [6,7,8], adipose stem cells [9,10], mesenchymal stem cells [11,12] and human induced pluripotent stem cells [13,14]. A common finding in such studies is the notable influence of vibration on the structural arrangement of F-actin filaments within the cytoskeleton.

Actin filaments are one of the components of the cytoskeleton, along with microtubules and intermediate filaments. Among these structures, actin filaments are the thinnest and have a crucial role in organizing the intracellular environment. They are involved in various critical mechanical cellular processes, including providing cellular adhesion, enabling mobility, determining cell shape, contributing to cellular strength, and maintaining cellular structure [15]. The dynamic nature of actin filaments enables them to respond rapidly to the surrounding microenvironment, much like a tensegrity structure, thereby imparting mechanical resilience to cells [16,17]. Additionally, the dynamic properties of actin filaments may play a crucial role in transmitting mechanical stress from the cell surface or membrane to the nucleus [18,19,20].

The cytoskeletal structures show temporal variability and complexity [21,22]. For instance, responses to external mechanical cues of actin filaments within a cell can fluctuate regularly every ∼24 h [23]. However, while most studies involving mechanical stimuli on cells are carried out using continuous stimuli over a specific amount of time, there have only been a handful of experimental studies exploring the temporal regularity of mechanical stimulation of cell cultures using various apparatus. Some of these studies include, for example, rhythmic stretching (a 12 h cycle of stretching at 1 Hz for three consecutive days) of cell cultures using a flexible silicon substrate [24]. Through this method, Rogers et al. [24] demonstrated clock genes (e.g., ARNTL, PER, and SOX) entrainment in adult stem cells, which is important because the entrainment can regulate circadian rhythms in the cells. Additionally, Vágó et al. [25] showed enhanced tissue homeostasis and histogenesis through rhythmic (1 h at the same time of the day for six days) uniaxial compression of chondrogenic cell cultures using biocompatible stainless plates pressing down from above the cells on a Petri dish. These results can have a valuable implication in tissue engineering since rhythmic (dynamic) mechanical stimuli can be used to optimize and improve the integration of tissue constructs within a host body [24]. Such *rhythmic* “ordered patterns in time” [26] studies could also be relevant in the dynamic cell culture system using a vibration system.

The potential impact of dynamic cell culture systems using vibration setups with controlled temporal variations remains largely unexplored. To address this gap, our study aimed to investigate the effects of rhythmic mechanical stimuli on cell cultures. We devised and carefully characterized an experimental setup that involved subjecting cell cultures to rhythmic vertical vibrations with varying pulse durations and intervals, allowing us to explore different stimulation parameters. In particular, we created three vibration conditions: Continuous Vibration (CV), Regular Pulse (RP), and Variable Pulse (VP). We categorize CV as non-rhythmic and RP and VP as rhythmic conditions, and No Vibration (NV) as a control condition. Departing from conventional non-rhythmic paradigms, we utilized principles from digital audio signal processing to generate rhythmic conditions by manipulating digital signals in various ways.

To measure the performance of the setup, we utilized several techniques. These included three-axis acceleration measurements to characterize the vibration regarding peak and Root Mean Square (RMS) acceleration. This enabled us to further estimate the mechanical movement in terms of RMS velocity and displacement of the vibrations.

In addition to vibration analysis, we measured fluid flow within the chamber of the cell culture plate while it was subjected to vibrations. Such fluid flow can result in shear stress at the cell level that can impact cellular mechanosensing mechanisms, including mechanosensitive ion channels, cell–extracellular matrix adhesions, cell–cell adhesions, and cytoskeleton organization [27]. Consequently, the activation of mechanotransduction pathways through shear stress can significantly influence various cellular behaviors such as alignment, gene expression, differentiation, and migration [27]. Recognizing the significance of fluid dynamics and shear stress and its impact on cellular behavior, we decided to employ particle tracking velocimetry (PTV) to analyze fluid flow. Among several non-contact measurement methods, such as particle image velocimetry (PIV) and laser Doppler velocimetry (LDV), we found that the PTV method is particularly well suited for our study involving a small confined space and microparticles. While PIV is commonly preferred for capturing larger flow fields at a fixed space and time, PTV allows us to track individual particles at any given spatial and temporal points [28]. This provides higher spatial resolution and can be advantageous for unsteady flow and temporal variation analysis, which is assumed under RP and/or VP conditions. The LDV technique can typically cover a range between a few millimeters to several meters per second in terms of velocity. This technique involves scattering of laser light and requires the particles to be sufficiently large for accurate measurement [29]. Therefore, it is not suited to measure microparticles. Consequently, this reinforced our decision to utilize the PTV method. We were able to estimate shear forces in close proximity to the fluid–solid interface where cells were attached, adding a novel insight and introducing an additional mechanical factor in vibration systems. This information is invaluable for comprehensively understanding the vibration apparatus.

To assess the impact of vibration on cellular function, we analyzed F-actin filament structure, focusing on changes in filament length, thickness, and angle. Additionally, we examined the cell cycle distribution as a secondary functional readout on the vibrated cells.

With these methodologies and measurements, we aimed to comprehensively evaluate the experimental setup and shed light on the potential functional implications on cellular responses to dynamic mechanical stimuli.

## 2. Materials and Methods

### 2.1. Cell Culture

HeLa cells (ATCC, CCL-2) were cultured in a growth medium (Dulbecco’s Modified Eagle Medium; DMEM) containing 10% fetal bovine serum and 1% penicillin–streptomycin in a cell culture flask (25 cm${}^{2}$) in an incubator at 37 ${}^{\circ }$C with 5% CO${}_{2}$. Before the plating of cells in 8-well glass-bottom slides (µ-slide, ibidi), the wells were coated with poly-L-lysine (Santa Cruz Biotechnology, Dallas, TX, USA). After the addition of poly-L-lysine (0.1% (*w*/*v*)), the slides were incubated at 37 ${}^{\circ }$C for five minutes, washed with phosphate-buffered saline (PBS) twice, and dried at room temperature for three hours, followed by disinfection under ultraviolet (UV) light for 15 min. Subsequently, trypsinized cells were added at 100,000 cells per well (growth area per chamber: 1 cm${}^{3}$) and were incubated for 24 h before treatment with vibration.

### 2.2. Experimental Design

#### 2.2.1. Vertical Vibration Setup

For vibration treatment, the ibidi glass-bottom slide (Figure 1a) was taken out of the incubator and placed and secured on top of a round (Ø = 100 mm) baseplate constructed by laser cutting a 4 mm thick PMMA sheet (Figure 1b) that is coupled with the vertical vibration generator (U56001, 3B Scientific) through a stand rod. The rod was connected to the speaker inside the device, and the frequency response of the generator ranged from 0 to 20 kHz. The generator was controlled by a digital signal generator developed in the Max/MSP environment (a visual computer programming language for music and multimedia, Cycling ‘74), where various experimental conditions (e.g., frequency, pulse durations, intervals, and waveform) were controlled and manipulated (a standalone application can be found in Appendix A). Through digital-to-analog conversion (DAC), an analog audio signal produced by the PC is sent to an arbitrary amplifier and eventually to the vibration generator, as illustrated in Figure 1c.

#### 2.2.2. Experimental Conditions

The cells were treated with vertical vibration using 50 Hz sine tone for 15 min under different conditions that varied in temporal regularity. The conditions were:No Vibration (NV): control (static) condition;Continuous Vibration (CV): continuous (uninterrupted) sound vibration;Regular Pulse (RP): rigidly regular pulses with a fixed pulse duration and interval at 1000 ms;Variable Pulse (VP): irregular pulses with pulse durations and intervals varying unpredictably between 1000 and 1500 ms

We categorized CV as non-rhythmic and RP and VP as rhythmic conditions. The VP condition was included to contrast as much as possible with the precise regularity observed in the CV and RP conditions. Three separate experiments were performed for each condition, using a new batch of cells each time.

#### 2.2.3. The Generation of Numerical Sequences for the Regular (RP) and Variable Pulse (VP) Conditions

The signals used to generate RP and VP were controlled and manipulated in the Max/MSP environment. For RP (Figure 2), a fixed input argument at 1000 ms was used for the “metronome object” to control signals. Accordingly, the pulse duration and interval were set at 1000 ms. For VP, a sequence of pseudorandom numbers was used (Figure 2), which was generated through the “random object” in Max/MSP. A specific sequence of random numbers can be recalled and regenerated using a seed message into the “random object”. When the seed message is not specified, the “random object” uses the system clock as the seed message. The generated numbers alter between 0 and 1, less than any integer argument. To create a sequence with a lower limit of 1000 and an upper limit of 1500, 500 was used as an integer argument for the “random object”. Then, the “plus object” was used to add 1000 to the output numbers of the “random object”, and the output of the “plus object” was used as an input argument into the metronome object. Consequently, the pulse duration and interval varied between 1000 and 1500 ms.

The generated sequences used for VP are stored separately and can be accessed and reused as necessary (Appendix A). One of the sequences is visualized in Figure 3 and was tested for its randomness using autocorrelation analysis (Figure 4) and was calculated using the runs test and entropy (Table 1). The blue shaded area of the autocorrelation in Figure 4 shows the confidence interval (α = 0.05), in which any value not significantly correlated to the previous value is shown. The runs test and entropy results are described and presented in Table 1.

### 2.3. Vibration Measurement

The vertical vibration setup was characterized using a 3-axis accelerometer (AX3, Axivity). An accelerometer is a type of sensor commonly used to capture various types of motion, including vibrations. The USB rechargeable accelerometer we employed was lightweight (16 g) and is widely used for research purposes. To ensure the vertical nature of the vibration, we opted to utilize a 3-axis accelerometer capable of measuring vibrations in three dimensions: X-axis, Y-axis, and Z-axis. This provides an advantage over traditional approaches that rely on single-axis sensors for measurement [30]. The accelerometer was secured on top of the cell culture plate, and the vibration was measured at a sampling rate of 3200 Hz.

### 2.4. Particle Tracking Velocimetry (PTV) Measurement

To further characterize the setup and to understand and estimate the fluid dynamics within the chambers of the cell culture plate, the fluid flow under different vibration conditions was characterized using the PTV setup as shown in Figure 5. It was possible to image the plane passing through the middle of the fluid in the glass cuvette that had the same 2D dimensions (10 × 10 mm) as the ibidi slides through the laser sheet created using an especially 3D-printed slit holder. We kept the volume of the fluid (DMEM cell media) the same (350 µL) as in the actual experimental setup. The camera (Grasshopper3 USB3 CCD, Teledyne FLIR) settings were:Shutter: 5 ms;Gain: 12 dB;Framerate: 25 FPS.

The “auto” settings of the camera were disabled.

In fluid mechanics, shear stress, denoted by τ, is a measure of the force per unit area that acts parallel to a surface due to the fluid’s velocity gradient. It is defined as:(1)$$\tau =\mu \frac{\partial u}{\partial y}$$
where μ, *u*, and *y* are dynamic viscosity, velocity, and height, respectively. When a fluid flows over a solid surface or boundary, such as at the bottom of the dish, it is commonly assumed that the fluid adheres to the surface without slipping, known as the no-slip condition [31]. This implies that the fluid velocity at the solid surface is assumed to be the same as the velocity of the solid surface. Assuming such a condition at the bottom of the dish, the shear stress experienced by the cells can be approximated as:(2)$$\tau \approx \mu \frac{u(y=y_{0})}{y_{0}}$$
where $y_{0}$ represents the first minimal height above the bottom surface where the fluid velocity can be reliably measured using the PTV technique.

### 2.5. Staining of F-Actin Filaments

Following treatment with vertical vibration, the cells were incubated for 15 min in the CO${}_{2}$ incubator at 37 ${}^{\circ }$C. Afterward, the cells were rinsed with cold PBS once and incubated with cold 4% paraformaldehyde (PFA) at room temperature for 15 min. Following this, the cells were washed twice with PBS and stored at +4 ${}^{\circ }$C. For staining of F-actin, fixed cells were permeabilized by incubation with 0.1% Triton X-100 in PBS at room temperature for 15 min. Next, the cells were incubated with AlexaFluor-488 Phalloidin (ThermoFisher, diluted 1:50 in 4% BSA/PBS) for one hour at room temperature and then washed three times with PBS to remove any unbound dye. Stained cells were stored at +4 ${}^{\circ }$C until microscopy analysis.

### 2.6. Microscopy and Image Acquisition

The ZEISS Elyra PS1 system was used with standard filter sets and laser lines with a Plan-APOCHROMAT, 63x/1.4 NA oil objective using the structured illumination microscopy (SIM) mode of the system. SIM imaging was performed using five grid rotations (0.51 mm grid) for 15 Z-planes with a 0.110 µm spacing. For SIM image reconstruction, the ZEN black software (MicroImaging, Carl Zeiss) was used with the following “method” parameters:Processing: manual;Noise filter: −4;SR frequency weighting: 1;Baseline cut, sectioning: 100/83/83;PSF: theoretical;Output: SR-SIM.

For image analysis, SIM images displayed as maximum intensity projections rendered from all Z-planes were used.

Nine different locations across the entire growth area of a well were imaged. Each vibration treatment was independently repeated three times, resulting in a total of 27 images per condition for analysis. All the acquired images can be found in Appendix A.

### 2.7. Image Analysis of F-Actin Filament Structures

The feature extraction of the acquired fluorescence images was performed using the Scikit and Scipy image packages in Python for further analysis. An “Actin Analyzer” algorithm was developed for this particular analysis and is available in Appendix A.

The algorithm depicted in Figure 6 was developed for this analysis to detect and extract the F-actin filament structural features. Filaments represented in the images were first segmented using a thresholding method on the output of a Laplacian filter (an edge-detecting method; step 2 of Figure 6) applied to the images. For the detection to be reliable even with images slightly out of focus, an adaptive threshold was used (step 3 of Figure 6), and pixels p(x,y) belonging to filaments were considered to follow the criteria:(3)$$\Delta p\left(x,y\right)\geq \langle \Delta p\rangle -{\mathrm{std}}_{x,y}\left(\Delta p\right)$$
where Δp(x,y), $\langle \Delta p\rangle$, and ${\mathrm{std}}_{x,y}\left(\Delta p\right)$ denote the value of the laplacian at position (x,y), the average of the laplacian over the whole image, and its standard deviation, respectively.

The found contours were then skeletonized (i.e., made one pixel thick), and extremely short contours were considered as noise and filtered out (steps 4 and 5 of Figure 6). Finally, slight gaps in the filaments were filled by connecting filaments whose ends were both at a distance shorter than a set value and of similar relative orientation (i.e., angle; step 6 of Figure 6). Additionally, the extreme outliers (over 100 µm) were filtered out in the post-data analysis.

### 2.8. Analysis of Cell Cycle Distribution

Cell cycle distribution was analyzed using the Click-iT Plus EdU Flow Cytometry Assay Kit from ThermoFisher, following the manufacturer’s instructions. Briefly, 24 h after the vibration treatment (in 8-well glass-bottom slides), cells were exposed to 1 mM of Click-iT-EdU for 2 h. After trypsinization, cells from two “8-wells” were collected in the same tube and fixed, permeabilized, and stained with Alexa 488 Click-it-EdU label reagent to detect newly synthesized DNA. Total DNA was stained by incubating cells in FxCycle PI/RNase staining solution (ThermoFisher, Waltham, MA, USA) for 30 min. The BD Accuri C6 Plus flow cytometer (Becton Dickinson, Franklin Lakes, NJ, USA) was used to quantify the percentage of cells in the G1, S, and G2/M cell cycle phases. Dot plots showing total DNA (PI stained) were analyzed using gatings shown in Appendix A. Four parallels were analyzed for each treatment.

### 2.9. Statistical Analysis

A t test was applied to compare the means of two independent data sets with normal distribution. For non-parametric data sets, Kruskal–Wallis (non-parametric ANOVA) was applied to test the significance level between experimental groups. As a post hoc test, the Wilcoxon rank-sum test was applied to compare the medians of the data sets with skewed distribution. All tests were performed using SciPy.stats package in Python. A *p* value < 0.05 was considered statistically significant. All the plots were generated in Python.

## 3. Results

### 3.1. Characterization of Vertical Vibration in Terms of Acceleration Parameters

The vertical vibration setup was characterized using a three-axis accelerometer (see Section 2). The measured acceleration in three dimensions (X, Y, and Z) showed acceleration only in the Z (vertical) direction in Figure 7, confirming that the direction of the vibration is vertical or perpendicular to the surface the setup rests on. A visual comparison in Figure 8 presents the differences between the experimental conditions. The fast Fourier transform (FFT) analysis revealed a few odd harmonics present above the fundamental frequency in the measured acceleration (Figure 9). The reason for the odd harmonics and the fundamental frequency being slightly higher (∼51 Hz) than the input frequency (50 Hz) could have been due to the mechanical resonances between the Ibidi cell culture slide and its lid. The total amount of time of vibration and silences (pulse intervals), the highest acceleration peaks, root-mean-square (RMS) accelerations, crest factors (the ratio between the highest peaks and the RMS values), RMS velocity (m/s), and RMS displacement (m) are presented in Figure 10. Additionally, the distribution of measured acceleration between the different experimental conditions is illustrated in Figure 11. The summary of the data sets is presented in Table 2.

### 3.2. Fluid Flow Patterns and Shear Stress Estimation in the Wells of the Cell Culture Plate through the Particle Tracking Velocimetry (PTV) Method

The PTV tracks revealed two vortexes in the plane where the direction of the flow was divided into left and right, resulting in negative and positive velocity and shear stress values at the bottom of the dish, respectively (Figure 12). Due to the symmetry of the glass cuvette, the flow pattern in that region is expected to be co-planar. The shear stress at the boundary layer (close to the bottom surface of the glass cuvette) was calculated using the velocimetry data measured from the images and is presented in Figure 12. Although the shear stress in non-constant flow observed under RP and VP is only an approximation and is not as reliable as in constant flow under CV, it was possible to visualize patterns with different regularities under RP and VP.

Under CV, due to the direction of the fluid flow that resulted in distinct vortexes, the estimated shear stress was the highest in the middle of each vortex close to the bottom surface of the cuvette. The shear stress varied more regularly under RP than VP. Furthermore, we analyzed the motion of the particles using an analyzer developed to extract motion over time [32]. The motiongram is a great tool to evaluate movement between successive frames in time, as presented in Figure 13. According to these estimations, the cells were exposed to more regular mechanical stress under RP than VP.

### 3.3. Impact of Vertical Vibration on the F-Actin Filaments in HeLa Cells

In order to determine the impact of vertical vibration on the cells, the F-actin filaments in cultures of adherent HeLa cells were examined. After subjecting the cells to a 15 min vibration treatment followed by 1 -min incubation without vibration, the cells were fixed, and the F-actin filaments were stained. We then used our custom algorithm (see Section 2) to measure the angle, length, and thickness of the F-actin filaments.

We report median values for the filament lengths and thicknesses here since the data sets show a non-normal (asymmetric) distribution (see Section 2). A detailed statistical summary of the data sets is available in Appendix A. In general, the data show a pattern of reduced F-actin filament size in all vibrated cell cultures, as presented in Figure 14a. In terms of filament lengths, treatment of all three vibration conditions produced shorter median filament lengths (CV: 0.60 µm, *p* < 0.001; RP: 0.64 µm, *p* < 0.001; VP: 0.64 µm, *p* < 0.001) than the control (NV: 0.68 µm). When it comes to filament thicknesses, compared to the control (NV: 0.100 µm), CV (0.095 µm, *p* < 0.001) and RP (0.097 µm, *p* < 0.001) resulted in thinner median F-actin filaments, while VP (0.101 µm, *p* < 0.001) produced slightly thicker filaments (Figure 14a). The impact of the vibration patterns on filament angle was also noticeable in the data. The filament angle data set showed a normal distribution; thus, t tests were employed to compare the differences in the means. Compared to the control (NV: 6.61 degrees), the mean angle of the filaments of the cell cultures was altered under CV (−1.67 degrees, *p* < 0.001), RP (1.05 degrees, *p* < 0.05), and VP (−0.47 degrees, n.s.) (Figure 14b).

### 3.4. Effect of Vertical Vibration on Cell Cycle Distribution

As a follow-up functional readout, we next analyzed if vertical vibration had an effect on the cell cycle distribution of the cells. The cell cycle analysis was performed 24 h after vibrational treatment using the EdU incorporation assay. As seen in Figure 15, the control (NV) exhibited a cell cycle distribution consisting of 44.88% in G1 phase, 45.02% in S phase, and 5.95% in G2/M phase. In CV-treated cells, a significant increase in G1-phase cells (47.23%, *p* < 0.001) and a significant decrease in G2/M-phase cells (5.08%, *p* < 0.01) was observed. RP-treated cells likewise displayed a significant increase in G1-phase cells (45.88%, *p* < 0.05), while VP treatment did not result in significant changes in the cell cycle distribution (Figure 15).

### 3.5. Correlations in Relation to the Magnitude of the Mechanical Stimuli

According to our data, the filament size (length and thickness) and accumulation of cells in the G1 phase exhibited a strong correlation with the mechanical parameters, while the filament angle appeared to be less/not influenced. The F-actin filaments of cells subjected to vibration exhibited a reduction in both length and thickness compared to the control (NV) cells. This effect was more pronounced in cells subjected to the non-rhythmic condition (CV), which had a higher RMS acceleration (Figure 10b) and shear stress compared to the rhythmic conditions (RP and VP) (Figure 12d), indicating a negative correlation between the magnitude of mechanical stimuli and the size of F-actin filament structure (Figure 16). Additionally, our data indicate that vibrated cells displayed a trend of G1-phase accumulation, with CV having a greater effect on cell cycle distribution than RP and VP. As a result, the observed increase in the level of G1-phase cells was positively correlated with the magnitude of the mechanical stimuli (Figure 16).

## 4. Discussion

The aim of this study was to develop a methodological protocol where cell cultures can be vertically vibrated rhythmically with varied temporal regularity (e.g., Continuous Vibration (CV), Regular Pulse (RP), and Variable Pulse (VP)). This was achieved by using a vertical vibration system and concept borrowed from music technology (i.e., the generation of rhythmic sound vibrations through audio signal processing in Max/MSP). The experimental setup was characterized in terms of acceleration (g), displacement (m), velocity (m/s), motiongram (quantity of motion), and estimated shear stress (Pa) in the cell culture chambers.

In our study, the rhythmic vibration conditions (RP and VP) translated into more dynamic movement with a varying degree of temporal regularity (rigid pulse duration and interval at 1000 ms or varied pulse duration and interval between 1000 and 1500 ms), lower RMS acceleration (RP: 0.97 g, VP: 1.03 g), and lower shear stress accumulation range (RP: 1.12–2.13 Pa, VP: 0.37–1.07 Pa) than the non-rhythmic condition (CV; RMS acceleration: 1.68 g; shear stress range: 3.18–3.53 Pa). Consequently, RP and VP produced mechanical stress that varied dynamically over the surface of the cell culture plate. CV also resulted in varied mechanical stress over the surface; however, the stress was constant and stable in comparison (Figure 12 and Figure 13).

To quantify the effects of vibration on the F-actin structure, an automated batch image analysis and feature extraction algorithm was developed to analyze the fluorescent F-actin images (“Actin Analyzer”; Appendix A). Batch processing was used to conduct an automated analysis of 108 images, and multiple tests were employed to verify the accuracy of the data produced by the algorithm (Appendix A). The effect of the three vibration conditions on the structure of F-actin filaments was evaluated by quantifying filament length, thickness, and angle.

The analysis revealed significant alterations in the structure of F-actin filaments in the vibrated HeLa cells compared to the non-vibrated cells and variations in the extent of changes observed in the results from CV, RP, and VP treatments. The main differentiating factor among the CV, RP, and VP conditions is the temporal regularity of the vibrations, which consequently affects the overall magnitude of the mechanical parameters. The significant negative correlation observed between shear stress and F-actin filament length, thickness, and angle (Figure 16) implies that, as the total accumulated vibration time and magnitude of mechanical stress increased, the filaments of the vibrated cells underwent a greater reduction in size. Actin filaments are known to be highly dynamic structures, regulated by both cellular mechanisms and mechanical forces [33], and several studies have reported that mechanical stress can cause actin filaments to shorten [34,35]. In addition, actin filaments can form a denser network under compressive loads, resulting in a shorter mean filament length [36]. However, when cells are subjected to shear stress, actin filaments can also be stretched [36,37]. Further characterization of the mechanisms underlying our observed changes in F-actin structure was beyond the scope of this study.

Mechanical forces can affect the progression of the cell cycle, as evidenced by previous studies that have reported contradictory findings. While some studies have demonstrated that mechanical stress can result in G1 cell cycle arrest [38,39], others have reported increased rates of progression through the G1 and G2 phases [40]. In our study, we observed an increase in the number of cells in the G1 phase of the cell cycle 24 h after CV and RP treatment (2.4% and 1.0% increases, respectively) (Figure 15).

To summarize, the main objective of this work was to introduce an experimental setup for exposing adherent cells to rhythmic vertical vibrations. While our focus was not solely on characterizing biological responses and underlying mechanisms, we sought to lay a solid foundation for future investigations. Various techniques were employed to characterize and quantify the mechanical forces exerted on the cells by such vibrations. By analyzing changes in F-actin structure and cell cycle distribution, we have provided examples of biological readouts that can be utilized to measure the functional effects of different vibration regimes. The described experimental system and characterization methodology can serve as a model for future studies investigating the impact of rhythmic mechanical stimulation on cells. We encourage subsequent investigations to delve deeper into the multifaceted nature of this phenomenon by considering critical variables such as amplitude, duration, frequency, and pattern of vibration to better comprehend the effect of rhythmic mechanical stimulation on important cellular features. Such features include cell proliferation, morphology, migration, gene and protein expression, signaling pathways, metabolic activity, and more. By exploring these variables, a deeper understanding of the specific cellular responses and underlying mechanisms in the context of rhythmic mechanical stimulation can be attained. Through our research, we aspired to not only illuminate the vast potential in *rhythmic* mechanical stimulation but also to catalyze further explorations that will unlock more profound insights into the intriguing interplay between mechanical forces and cellular behavior. This contribution to the field promises to pave the way for transformative discoveries in understanding cellular responses to mechanical stimulation.

## Figures and Tables

**Figure 1 bioengineering-10-00811-f001:**
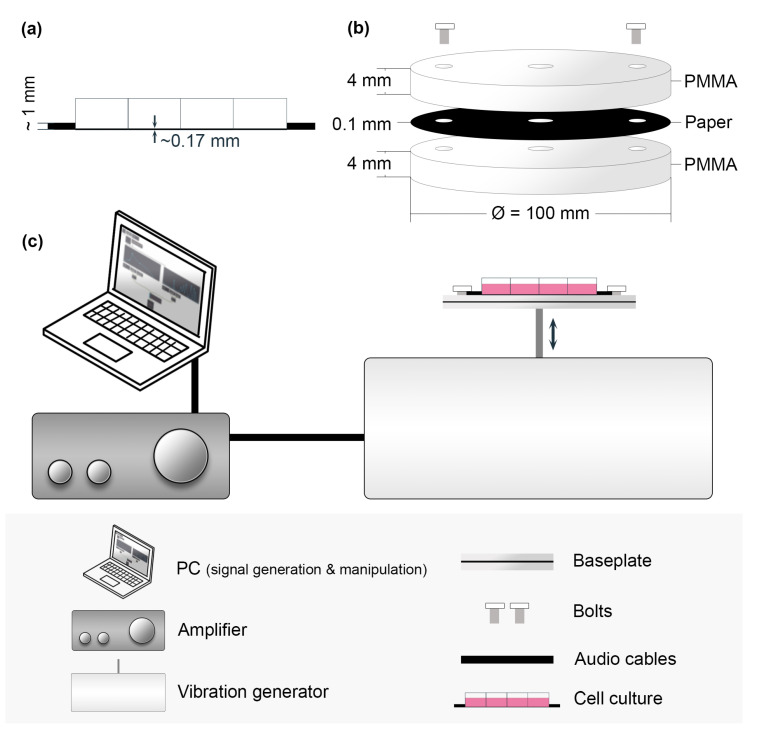
A schematic diagram of the vertical vibration setup and its components. (**a**) Side view of the thin 8-well glass-bottom slide. (**b**) Exploded view of the PMMA baseplate. The black paper is included for better visualization, and the paper is not expected to impact the vibration propagation. (**c**) The vertical vibration experimental setup, including a PC, amplifier, and vibration generator.

**Figure 2 bioengineering-10-00811-f002:**
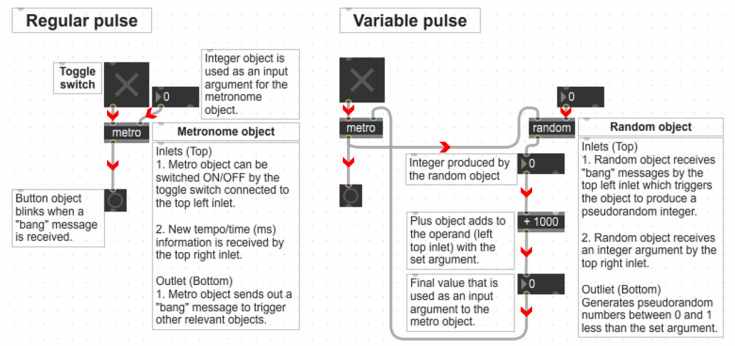
A section from the Max/MSP patch illustrating how the numerical sequences were generated to manage the digital audio signals regularly or irregularly within the Max/MSP environment. The red arrowheads were added to indicate the message or signal flow. The metronome object and a fixed integer argument ( i.e., 1000) are used to vary the signals regularly for the Regular Pulse (RP) condition. The random object that generates a sequence of pseudorandom numbers from 0 and 1 less than a set argument is used as input arguments into the metronome object to vary the signals irregularly for the Variable Pulse (VP) condition.

**Figure 3 bioengineering-10-00811-f003:**
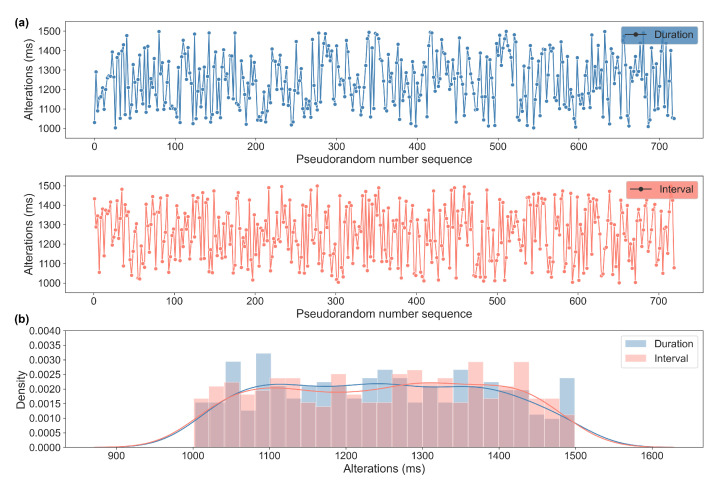
(**a**) Visual representations of a pseudorandom number sequence used for the Variable Pulse durations and intervals. (**b**) A density histogram analysis shows the distribution of the durations and intervals. The two colored lines, the kernel density estimations, illustrate an approximately uniform distribution of the series of numbers within the set range between 1000 and 1500.

**Figure 4 bioengineering-10-00811-f004:**
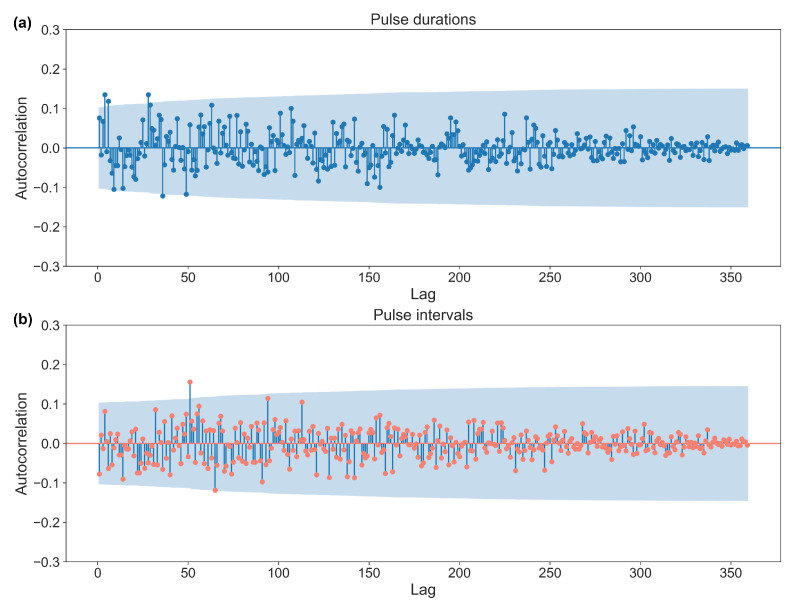
The autocorrelation analysis of a pseudorandom number sequence used for the varied (**a**) pulse durations and (**b**) pulse intervals. The maximum lag is 360 in this data set. Any data point falling within the blue shaded area (α = 0.05) indicates that it is not significantly correlated with the previous point. The analysis illustrates that the coefficient values of the autocorrelation are close to 0, and the consecutive numbers are mostly not significantly related in the sequence. Therefore, the sequence is most likely produced randomly.

**Figure 5 bioengineering-10-00811-f005:**
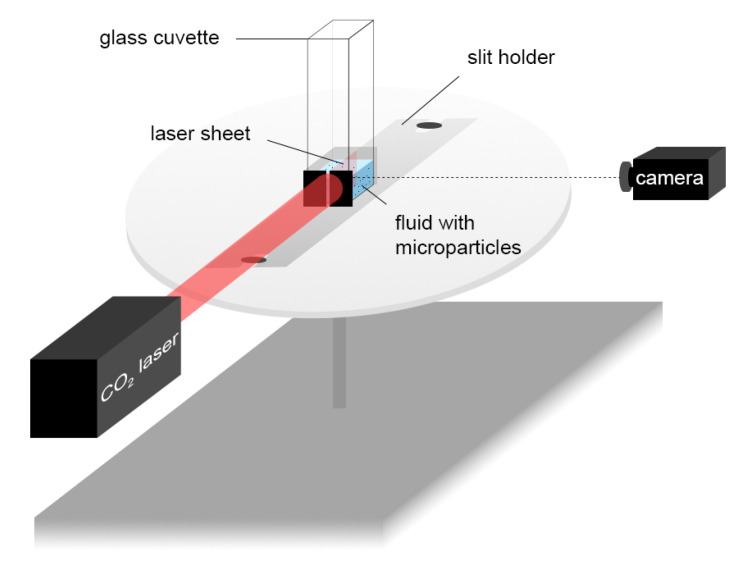
Particle Tracking Velocimetry (PTV) setup with a VWR glass cuvette (10 × 10 mm, the same dimensions as a well of the ibidi glass bottom 8-well slide) secured on the 3D-printed slit holder on the vertical vibration generator. The slit holder and glass cuvette were held down firmly on the baseplate with screws. The laser sheet in the cuvette was created using a CO${}_{2}$ laser and the slit, which was 0.1 mm in width and 10 mm in height. The glass cuvette was filled with DMEM cell media (350 µL) and microparticles (Ø = 10 µm). A Grasshopper3 USB3 CCD camera was used to capture images. The images were taken using a stroboscopic technique by keeping the framerate of the camera at 25 FPS while the vibration generator produced a 50 Hz sine tone.

**Figure 6 bioengineering-10-00811-f006:**
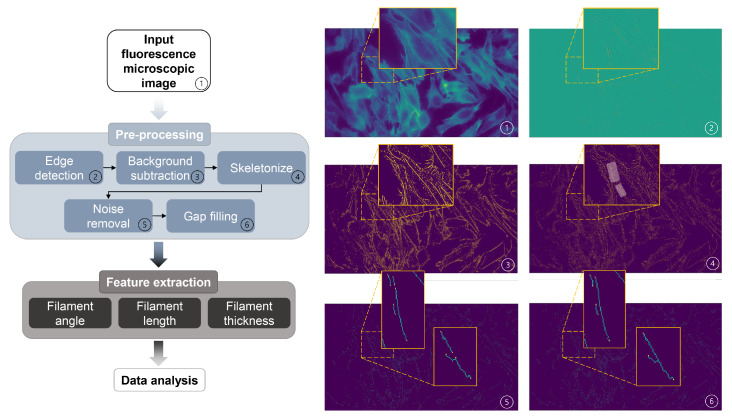
Schematic diagram of the image feature extraction algorithm to analyze F-actin filament structures (filament angle, length, and thickness) using fluorescence microscopic images. The two shaded areas in image 4 indicate the zoomed-in positions shown in images 5 and 6. (**1**) Acquired fluorescence microscopic image used as input; (**2**)–(**6**) images filtered through specific steps, and extracted features (filament angle, length, and thickness) were exported for further data analysis.

**Figure 7 bioengineering-10-00811-f007:**
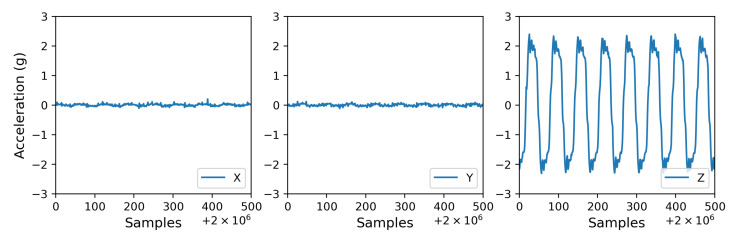
A comparison of the acceleration measurement of the vertical vibration setup between X, Y, and Z dimensions shows acceleration only in the Z (vertical) direction.

**Figure 8 bioengineering-10-00811-f008:**
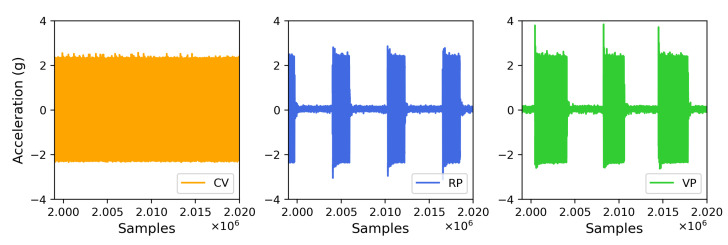
A comparison of measured acceleration between the experimental conditions. Note the continuous (uninterrupted) acceleration measured under Continuous Vibration (CV), rigidly the same pulse duration and interval under Regular Pulse (RP), and varying pulse duration and intervals under Variable Pulse (VP).

**Figure 9 bioengineering-10-00811-f009:**
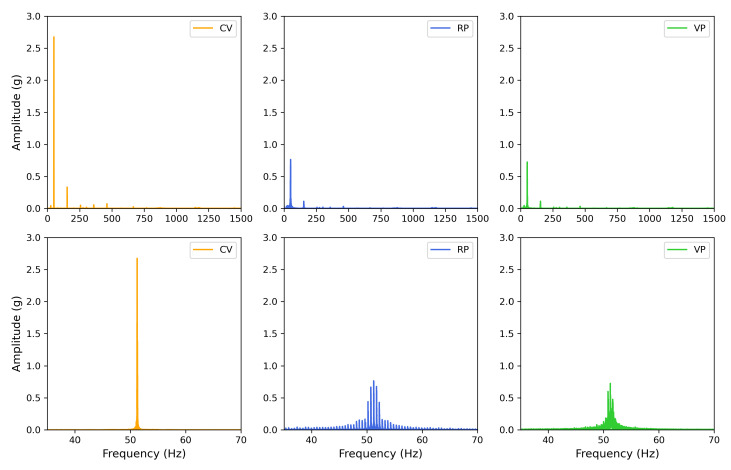
Fast Fourier Transform (FFT) analysis of the acceleration measurement of each experimental condition. (**Top row**) The full range of the FFT analysis shows a few minor peaks around the odd harmonics above the fundamental frequency. (**Bottom row**) The same FFT analysis zoomed in around 50 Hz to show the fundamental frequency (∼51 Hz) in the measured acceleration.

**Figure 10 bioengineering-10-00811-f010:**
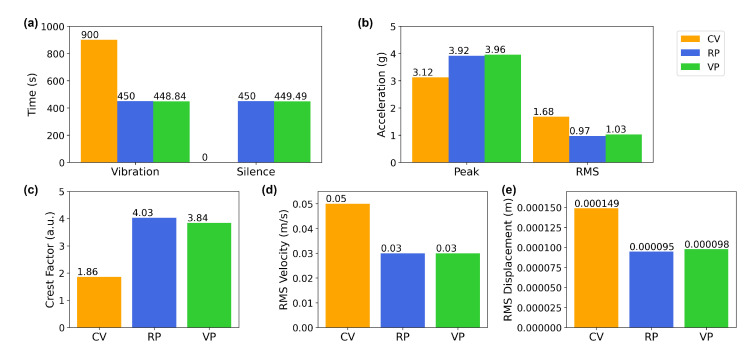
Visual representation of the parameters between the different vibration conditions. (**a**) A comparison of the total vibration and silence times between the conditions. (**b**) A comparison of the highest acceleration peak and RMS acceleration. (**c**) The crest factors (the ratio between the peak and RMS) show how “peaky” the signal is. The higher the value, the more the peakiness. CV scores show a lesser crest factor than the other two conditions, meaning the acceleration was more stable, and there was less variation between the acceleration peaks and the RMS throughout the sampled period. This corresponds to the RMS acceleration value, where CV scored the highest RMS value. (**d**) Comparison of the RMS velocity and (**e**) RMS displacement between different conditions.

**Figure 11 bioengineering-10-00811-f011:**
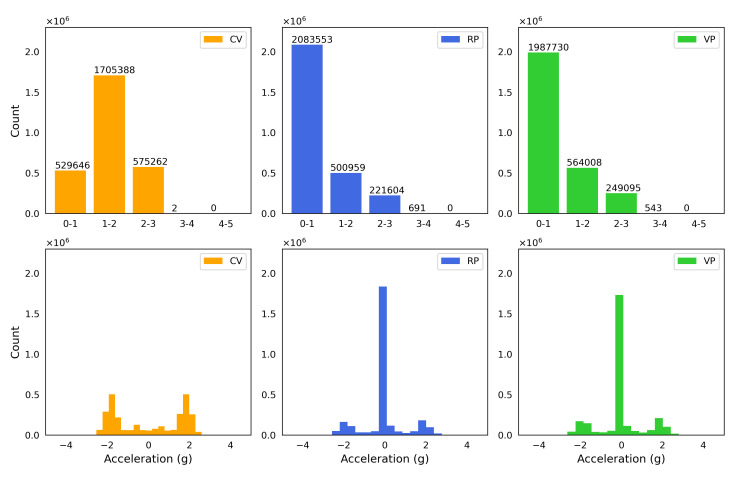
Distribution of the acceleration of the vertical vibration under each condition. Under RP and VP, the distribution is dominated by the lowest category (0-1) in (**Top row**) and around 0 g in (**Bottom row**), which represent the intervals (silences) between pulses. (**Top row**) Sub-divisions of absolute values of the measured acceleration. (**Bottom row**) Histograms of the measured acceleration.

**Figure 12 bioengineering-10-00811-f012:**
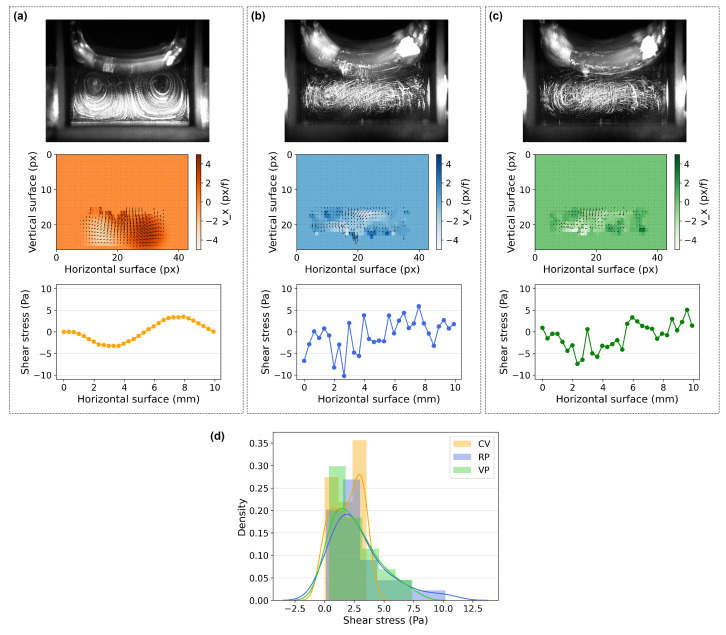
PTV results. The maximum intensity projection images, velocity fields, and shear stress plot at the boundary layer close to the bottom surface of the cuvette are shown according to the experimental conditions (**a**) CV, (**b**) RP, and (**c**) VP. (**d**) Distribution of estimated shear stress in absolute values, which shows a smoothed out distribution and where the values are concentrated.

**Figure 13 bioengineering-10-00811-f013:**
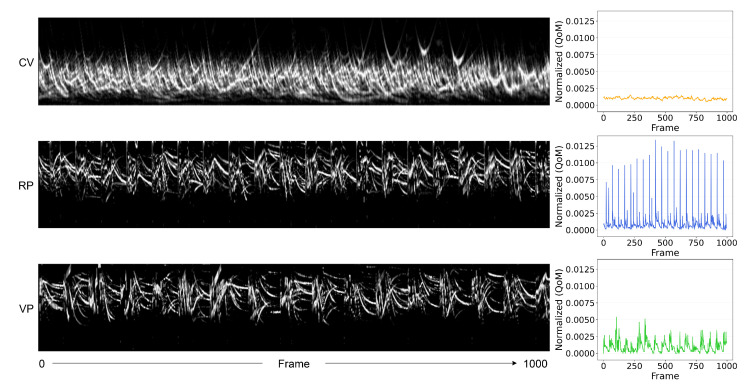
Motion analysis of the particles from the images obtained using the PTV setup (Figure 5). The motiongrams (in black background) and the quantity of motion (QoM) plotted over time (1000 frames) reveal the contrasting degrees of regularity in the motion of the particles between CV, RP, and VP conditions.

**Figure 14 bioengineering-10-00811-f014:**
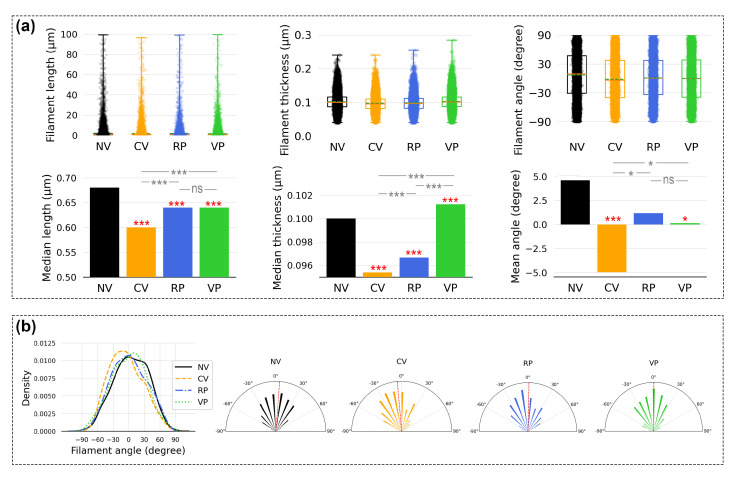
Analysis of the extracted features from the microscopic images. The results are statistically compared between the experimental conditions (NV, CV, RP, and VP). Red stars indicate the test that compared one of the vibration conditions to NV (control), gray stars indicate the test that compared the in-between vibration conditions, and ns indicates not significant. (**a**) The boxplot analysis (**top**) shows the distribution of the data. The box represents 50% of the data points, the green dotted line in the box represents the mean, and the orange line represents the median. The bar plots (**bottom**) show the differences in median filament lengths, thicknesses, and mean filament angles. (**b**) Filament angle distribution is presented with a density histogram and polar plots. Note that the angle does not show directions except for the 2D orientation within the collected microscopic images. The red dotted line in the polar plots indicates the mean value. * *p* < 0.05, *** *p* < 0.001.

**Figure 15 bioengineering-10-00811-f015:**
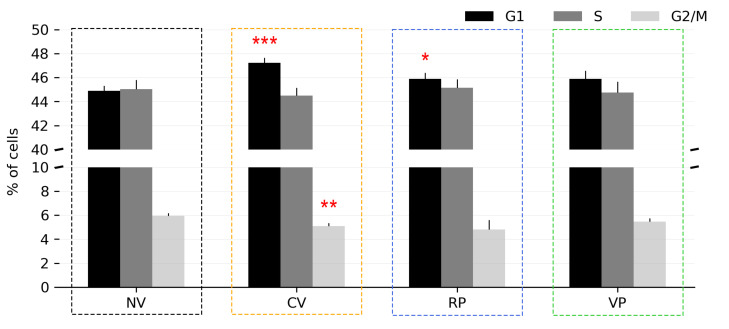
Cell cycle distribution in control (NV) and vibrated (CV, RP, VP) HeLa cells analyzed 24 h after vibration treatment. Red stars indicate a significant difference relative to the control (NV) cells. Flow cytometry dot plots showing the gating strategy are included in Appendix A. * *p* < 0.05, ** *p* < 0.01, *** *p* < 0.001.

**Figure 16 bioengineering-10-00811-f016:**
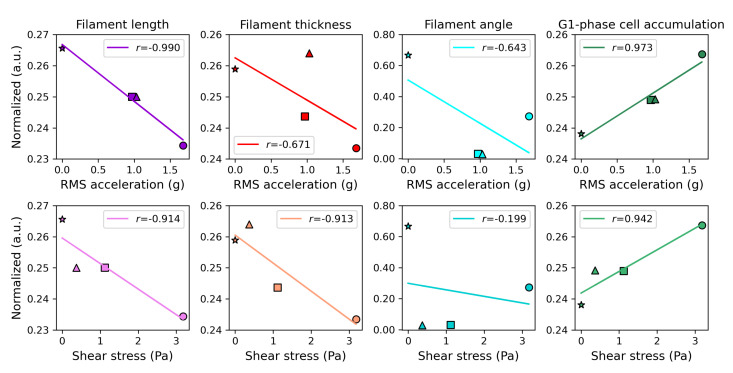
Correlation between the extracted features from the microscopic images (median filament length and thickness and mean filament angle) and characterized mechanical parameters (RMS acceleration and estimated shear stress). *r* is the Pearson correlation coefficient. Markers: NV—star, CV—circle, RP—square, VP—triangle.

**Table 1 bioengineering-10-00811-t001:** Randomness tests of the pulse durations and intervals used to generate the Variable Pulse (VP) condition. The total amount of numbers (n) was 360 for each sequence. The *p* value of the runs test indicates that if *p* < 0.05, then the sequence was not produced randomly. The entropy values are computed using Python’s Scipy.stats.entropy function. The closer the entropy value is to the maximum value, the higher the randomness of the sequence of numbers.

	n	Runs Test (*p* Value)	Entropy Max Value	Entropy Value
Pulse duration	360	0.4	5.89	5.27
Pulse interval	360	0.1		5.27

**Table 2 bioengineering-10-00811-t002:** A comparison of parameters between the different vibration conditions.

Data Set	CV	RP	VP
Input frequency (Hz)	50	50	50
Measured fundamental frequency (Hz)	∼51	∼51	∼51
Total vibration time (s)	900	450	448.84
Total silence (intervals) time (s)	0	450	451.16
Acceleration peak (g)	3.12	3.92	3.96
Acceleration RMS (g)	1.69	0.97	1.03
Crest factor	1.86	4.03	3.84
Velocity RMS (m/s)	0.049	0.029	0.031
Displacement RMS (m)	0.00015	0.0001	0.0001

## Data Availability

Data can be found in the Appendix A.

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
