# Peer review of "Characterization of Mechanical and Cellular Effects of Rhythmic Vertical Vibrations on Adherent Cell Cultures"

_bioengineering, 2023, doi:10.3390/bioengineering10070811_

Round 1
Reviewer 1 Report
A) General remarks
The research presents in this paper a very interesting topic, as well as results that are of wider significance when it comes to mechanical characterisation. The paper is concise and clear. The literature in the paper is adequately cited.
1. In the case of literature, all the cited references are relevant to the research. No unnecessary self-citations were detected.
2. The English language used is without major mistakes and no significant connections are needed.
3. The abstract is well written. The role of the abstract is to give a basic overview of the paper. In this case, the abstract gives a good introduction to the paper without specific data and is very informative even for those none-familiar with the topic reader. However, the novelty aspects of the paper must be presented stronger. Additionally, for clarity of the abstract, it is suggested not to use abbreviations but full names. The abbreviations are to be introduced the first time they occur in the text but not in the abstract.
4. The introduction is mostly well-written and follows all the rules of the proper instruction on the topic. However,
a. The last paragraph where the aim scope and novelty is presented must be improved. The aim has to be presented clearly. The novelty in the current version is not presented. This must be improved.
b. Introduction, Material and methods or information on experimental setup do not give a proper introduction to vibration measurements, especially none contact methods. Thus it is not known why the authors have chosen PTV. Some alternative techniques must be introduced and explained why PTV was chosen. Example Particle Image Correlation can be used for eg PIV measurement and CFD simulations of an air terminal device with a dynamically adapting geometry to track particles in the air but not their vibrations (although this can be also done indirectly). 3D Laser Doppler Vibrometry can measure the vibration of both small and big structures and is especially lightweight (e.g. DOI: 10.3390/s23031263) but not below 1mm2 (so no microparticles). Additionally, with some expertise, it can be used to measure the vibration of objects in the fluid. Thus PTV is ideal for the case presented. In general, there is no introduction in the paper to vibration measurement techniques. Strongly suggest improving and evaluating possible techniques of non-contact measurements.
5. Some small editorial problems with the values and unit presentation. It is customary to make a space between the value and the unit, except deg C and %. Some errors can be found in the article e.g. with â—¦C and %
6. The paper does provide proper discussion but limited conclusions. Again the novelty aspect must be pointed out strongly.
B) Item remarks
Fig .3 caption and text are hardly visible. Please improve and unify so Fig.3 and 4 look the same.
Fig.6 Schematic diagram texts are hardly visible.
Fig.12 is too small. Nothing is visible. Suggest leaving a, b, and c in one row and d moved (and enlarging) to the second row.
C) Conclusions:
The biggest problem of the article is also the clear presentation of the novelty of the research topic and the lack of evaluation of state-of-the-art cases of vibration measurement techniques.
However, the article is clear without many problems with methodology. Thus, the reviewer suggests the minor corrections mentioned previously and asks the authors to answer the fundamental questions- how the article can advance this field of study?
Author Response
Thank you very much for the opportunity to submit a revised manuscript “Characterization of Mechanical and Cellular Effects of Rhythmic Vertical Vibrations on Adherent Cell Cultures” for publication in Bioengineering. We appreciate the valuable comments and advice from the reviewers; they have helped to improve our work.
We have gone through the comments and considered them with great care. After a series of discussions, we have decided to remove and/or add several sections in the manuscript in response to the comments. The language of the manuscript has also been revised throughout the text. Our responses are shown in bold below with an indication of what was changed with line numbers. Additionally, we are submitting a revised manuscript.

Reviewer 2 Report
This paper could be extremely interesting if results were presented on a less "validation" base and more on a biological base.
First of all, it not clear why and how PMMA and paper are combined together and what does this system cause on the application of a well plate for cell culture (also to understand if there is a possible signal filtration or modification). Fig 1 helps but is not clear, how is everything assembled?
Why choosing this king of randomness of the signal? is it a physiological explanation or interpretation of a phenomena?
The ptv is very useful for the authors to validate their model, although, it cannot be valid for a biological interpretation of the results. If so, a scientific explanation of this relation should be widely presented and demonstrated.
This reviewer understand the importance of imaging especially for proteins filaments etc, but the quantification of the cell vitality, percentage and metabolic activity must be presented to validate such a system. Cells can resist and show their filaments for days without proliferating so imaging cannot be an index of biomedical applications reliability.
A complete explanation on how the shear stress (and where) has been evaluated must be inserted.
Fig. 2b can be totally avoided, does not have scientific value.
Cell cycle dependance on vibration state? Really, why? Assays are always useful but it depend on how they have been scientifically used.
Fig 16 is very useful to understand the statistical significance of these measurements but it also shows that the data are not reliable, is it a matter of the measurement system or the experimental results themselves?
Please provide other quantifications of the cell state.
Supplementary files require OSLO university account
N/A
Author Response

(The authors gave the same response as above.)

Round 2
Reviewer 2 Report
N/A